# Combination of High-Resolution Structures for the B Cell Receptor and Co-Receptors Provides an Understanding of Their Interactions with Therapeutic Antibodies

**DOI:** 10.3390/cancers15112881

**Published:** 2023-05-23

**Authors:** Puja Bhattacharyya, Richard I. Christopherson, Kristen K. Skarratt, Jake Z. Chen, Thomas Balle, Stephen J. Fuller

**Affiliations:** 1Sydney Medical School Nepean, Faculty of Medicine and Health, The University of Sydney, Kingswood, NSW 2750, Australia; 2Blacktown Hospital, Blacktown, NSW 2148, Australia; 3School of Life and Environmental Sciences, The University of Sydney, Sydney, NSW 2006, Australia; 4Nepean Hospital, Kingswood, NSW 2747, Australia; 5Sydney Pharmacy School, Faculty of Medicine and Health, The University of Sydney, Sydney, NSW 2006, Australia; 6Brain and Mind Centre, The University of Sydney, Camperdown, NSW 2050, Australia

**Keywords:** B cell, B cell receptor, CD19, CD81, CD22, cryo-electron microscopy, crystallography, monoclonal antibody

## Abstract

**Simple Summary:**

The treatment of B cell malignancies was transformed by the development of a monoclonal antibody—rituximab—that targets CD20, a protein expressed on the surface of B cells. However, some types of B cell malignancies do not express CD20 or can reduce the expression of the protein to escape therapy. Consequently, there is a need to develop tailored therapies against other targets expressed by B cells. The structures of key B cell proteins, the B cell receptor, and its co-receptors CD22, CD19 and CD81, have recently been solved and provide the opportunity to guide development of new antibodies and other therapies targeted at malignant B cells. Here, we review high-resolution protein structures of the BCR, CD22, CD19 and CD81 molecules, treatments that have been developed against these targets and discuss structural features that will enable the design of novel antibodies.

**Abstract:**

B cells are central to the adaptive immune response, providing long lasting immunity after infection. B cell activation is mediated by a cell surface B cell receptor (BCR) following recognition of an antigen. BCR signaling is modulated by several co-receptors including CD22 and a complex that contains CD19 and CD81. Aberrant signaling through the BCR and co-receptors promotes the pathogenesis of several B cell malignancies and autoimmune diseases. Treatment of these diseases has been revolutionized by the development of monoclonal antibodies that bind to B cell surface antigens, including the BCR and its co-receptors. However, malignant B cells can escape targeting by several mechanisms and until recently, rational design of antibodies has been limited by the lack of high-resolution structures of the BCR and its co-receptors. Herein we review recently determined cryo-electron microscopy (cryo-EM) and crystal structures of the BCR, CD22, CD19 and CD81 molecules. These structures provide further understanding of the mechanisms of current antibody therapies and provide scaffolds for development of engineered antibodies for treatment of B cell malignancies and autoimmune diseases.

## 1. Introduction

Activation of the B cell receptor (BCR) determines the fate of B cells at several stages of their life. In bone marrow, development and selection of pre-B cells is determined by the strength and timing of BCR signaling, and maturation to naïve B cells relies on tonic signaling. In the lymph nodes, binding of antigen to the BCR directs activation, proliferation and differentiation of B cells into antibody secreting plasma cells or memory B cells that provide long-lasting immunity after infection [1]. Signaling through the BCR contributes to the introduction of somatic hypermutations in the variable regions of immunoglobulin genes and heavy-chain class switching events to generate higher-affinity BCRs in memory B cells. In addition to activating the B cell to generate antigen-specific antibodies for humoral immunity, the BCR contributes to cell-mediated immunity by internalizing antigen for processing and presenting to T cells that generate cell-mediated immunity [2].

Considering the role of BCR signaling for survival, differentiation and proliferation in normal B cells, it is to be expected that signaling plays a critical role in transformation and survival of malignant B cells [3]. In addition, several proteins have been identified that interact with and modulate BCR signaling (Figure 1) [4,5]. A co-receptor complex that includes CD19 and CD81, and complement receptor type 2 (CD21), augments BCR signaling whereas CD22 is an inhibitory co-receptor [6,7]. A role for antigenic stimulation of the BCR in the pathogenesis of B cell malignancies has been hypothesized. Hepatitis C virus (HCV) infection is linked to the development of splenic marginal zone lymphoma (SMZL) and treatment of patients to clear HCV infection eliminates the lymphoma in most cases. Similarly, Helicobacter pylori infection is associated with gastric mucosa-associated lymphoid tissue (MALT) lymphomas and antibiotic treatment often leads to remission. In addition to these examples of antigen-dependent development of B cell lymphomas, antigen-independent, tonic BCR signaling has been described in chronic lymphocytic leukemia (CLL) and subtypes of diffuse large B cell lymphoma (DLBCL) [8].

The development of therapies targeted at specific proteins (e.g., CD antigens) on the surface or within malignant B cells has revolutionized the treatment of B cell cancers. The two main types of drugs are small molecule inhibitors and engineered antibodies. Small molecule inhibitors are designed to inactivate kinases and interrupt signaling pathways, such as the BCR signaling pathway, which are dysregulated in B cell cancers [9]. Monoclonal antibodies bind to known B cell surface antigens and induce cell death through several mechanisms [10]. Conjugated antibodies are internalized after binding to their target and lead to cell death by releasing a cytotoxin or radioisotope whereas unconjugated antibodies bind directly to the cell surface antigen and mediate apoptosis through antibody-dependent cellular cytotoxicity (ADCC), complement-dependent cytotoxicity (CDC), antibody dependent cellular phagocytosis and direct cell killing [11]. Bispecific antibodies engage two different target antigens simultaneously to direct immune specific effector cells into close contact with tumor cells and consist of variable domains linked together to form a single-chain antibody. These include bispecific T-cell-engager antibodies, dual affinity re-targeting antibodies and tandem single-chain variable fragments [12,13].

The treatment of B cell malignancies using small molecule inhibitors, and monoclonal antibodies targeting CD20, a well-established B cell target, has been extensively reviewed [14,15]. However, this review will focus on the recently published cryo-EM structures for murine and human BCRs, the BCR co-receptor, CD19, bound to CD81 [16,17,18,19], and the crystal structure of CD22 [20]. Monoclonal antibodies that have been developed against known antigen targets, and potential targets revealed from their structures.

## 2. The B Cell Receptor

### 2.1. B Cell Receptor Structure

BCRs consist of an antigen recognition module composed of one of five surface membrane forms, or isotypes, of immunoglobulin (mIgM, mIgD, mIgG, mIgA or mIgE) bound to two transmembrane signaling subunits, Igα (CD79a) and Igβ (CD79b), that form a heterodimer [16,17]. Each mIg is made up of two light and two heavy polypeptide chains connected by disulfide bonds that form a “Y” shape with two antigen binding (F_ab_ domain) “arms” connected to a “leg” (F_c_ domain) that is inserted into the surface membrane and interacts with the CD79a/CD79b heterodimer [16,17]. The F_ab_ arms contain a variable (V) region that binds to antigen, and a complex process of random recombination of V region genes results in an enormous range of antigen binding specificities that are unique to individual BCRs [21]. BCRs are clonally distributed on B cells resulting in sub-populations that express receptors targeting specific antigens. Binding of antigen to the V region of the F_ab_ induces signal transduction to the interior of the B cell via the CD79a/CD79b heterodimer. The first step of signal transduction to the nucleus is phosphorylation of intracellular immunoreceptor tyrosine-based activation motifs (ITAMs) in the CD79a and CD79b heterodimers by the SRC family of protein kinases, Lck/Yes-related novel (Lyn) tyrosine kinase and Fyn, B-lymphoid kinase (Blk) or spleen tyrosine kinase (Syk) [22,23]. The phosphorylated tyrosine residues then act as docking sites where a higher order signaling complex is assembled that gives rise to B cell activation, maturation and generation of immune responses [16,24]. This process requires second signals, many of which are provided by CD4^+^ helper T cells. In lymph nodes, dendritic cells present antigen bound to MHC class II molecules to naïve T cells that differentiate into T follicular helper (T_FH_) cells and move to the border of B cell follicles [25]. In these follicles, the BCR binds the same antigen, which is internalized, processed and presented on the cell surface on MHC class II molecules. The B cell presents the antigen to T_FH_ cells that are stimulated to express CD40 ligand, which ligates CD40 on the B cell, stimulating B cell proliferation and differentiation [26]. The T_FH_ cells also secrete cytokines required for Ig isotype switching [27], and cognate interactions with B cells induce proliferation, somatic hypermutation and antibody affinity maturation. These processes ultimately generate antibody-secreting plasma cells and memory B cells. BCR signaling can also occur independently of antigen binding by tonic activation of the receptor. Tonic BCR signaling is essential for the survival of B cells at different stages of their development and is mediated by low-level constitutive signaling via phosphoinositide 3-kinase (PI3K) [28]. A co-receptor complex that includes CD19 and CD81, as well as complement receptor type 2 (CD21) [29], augments BCR signaling by lowering the threshold for activation (Figure 1) [6]. In contrast, CD22 is an inhibitory co-receptor that diffuses laterally in the surface membrane to dampen BCR signaling both in the resting state and following activation [7]. Immature B cells in the peripheral blood and lymphoid organs express both mIgM and mIgD, while in contrast, memory B cells express mostly mIgG that shows enhanced responses to antigen compared to mIgM and/or mIgD. This suggests the IgG BCR transmits a distinct signal compared to the IgM or IgD-BCR [30].

In the past, it has been difficult to obtain high-resolution structural information on the intact BCR, a transmembrane receptor with a characteristically flexible hinge region in the heavy chain that links the extracellular Fab region to the Fc region. Furthermore, critical interactions between mIg and CD79a/Cd79b are located within the plasma membrane that to date have been unable to be visualized by crystallography. However, three groups have recently determined the full-length transmembrane structures of the BCR using cryo-EM [16,17,19]. These studies of the murine IgM BCR [19], the human IgG BCR and IgM BCR [16], and a second study of the human IgM BCR (Figure 2) [17], confirmed previous biochemical, fluorescence resonance energy transfer, computational modelling and experimentally guided modelling [31,32,33], showing the F_ab_ arms in a Y shape free of the CD79a/CD79b heterodimer bound to the mIg F_c_ domain with 1:1 stoichiometry [34]. The CD79a/CD79b heterodimer consists of an extracellular domain, a linker and a transmembrane domain. The extracellular domains (ECDs) of both proteins have a similar conformation and the two proteins interact via both the ECD and transmembrane domain ™. In the ECD, a disulfide bond is formed between Cys119 of CD79a and Cys136 of CD79b. The ECD interface is further supported by π-π stacking of aromatic rings, and hydrogen bonding. There are extensive hydrogen bonds and van der Waals interactions across the TM interface that is conserved across species. The linker peptide (Lµ_B_) passes through an “O-ring” formed between CD79a and CD79b that introduces the ECD into the plasma membrane [16,17], where the TM helices form a compact bundle dominated by van der Waals forces and several hydrogen bonds [16,17,19]. Connecting the IgM linker region to CD79a/CD79b though a space in the heterodimer supports the theory that the BCR is assembled in the endoplasmic reticulum [31]. The cytoplasmic tails of the BCR subunits were not identified, suggesting that they are structurally flexible and difficult to characterize using cryo-EM.

All 5 antibody isotypes form distinct BCRs that fine-tune the B cell response to antigens [36]. Differences in IgG- and IgM-BCR structure are likely to contribute to different sensitivities and thresholds to antigen activation [37,38]. The IgG molecule is made up of two light chains, each with one V and one L domain, and two heavy chains each with one V domain and three C domains and a spacer hinge region between the first and second C domains, whereas the IgM heavy chains have 4 constant domains [39]. The cryo-EM structures of the IgM- and IgG-BCR are markedly different, with mIgM located closer to the membrane and interacting with CD79a/CD79b on its side, whereas mIgG sits on top of CD79a/CD79b. The different assemblies of IgG-BCR (“head to tail”) and IgM-BCR (“side-by-side”) may explain the observed differences in their signaling [40]. Both heavy chains interact with CD79a/CD79b in the IgG-BCR, and only one IgM-BCR heavy chain interfaces with its CD79a/CD79b heterodimer.

### 2.2. Targeting the BCR for Treating B Cell Malignancies

A successful therapeutic monoclonal antibody has high specificity for its B cell target and binds to an antigen only expressed by the target of interest, decreasing the risks of off-target effects and toxicities. CD79b is an excellent target for immunotherapy of B cell malignancies. CD79b is expressed in early B cell development, before immunoglobulin gene rearrangement and CD20 expression, and is found on B cell lymphomas, most precursor B cell acute leukemias. CD79b expression is limited to the B cell lineage unlike CD79a, which has been reported to be positive in acute myeloid leukemias and normal megakaryocytes, although positivity appears to be dependent on the antibody clone [41]. In addition, although cell surface CD79b cannot be detected on CD14+ monocytes, T cells and CD34+ hematopoietic precursor cells, these cells express aberrant CD79b mRNA that makes CD79b unsuitable for T cell receptor-based therapies [42]. However, with almost exclusive cell surface expression on B cells, and as an essential component for BCR expression and function [1], CD79b is an attractive target for chimeric antigen receptor (CAR) and monoclonal antibody therapies.

### 2.3. Polatuzumab Vedotin

Polatuzumab vedotin (polatuzumab vedotin-piiq; Polivy™, Roche, Basel, Switzerland) is a CD79b-directed antibody covalently linked to the anti-mitotic cytotoxic agent monomethyl auristatin (MMAE) that disrupts microtubules, inducing cell cycle arrest and apoptosis [43]. The parental antibody for polatuzumab is the anti-human CD79b antibody, SN8 [44]. The SN8 monoclonal antibody (mAb), and SN8a and SN8b mAbs, were generated using a B prolymphocytic leukemia (PLL) antigen preparation to immunize mice. The epitope targeted by SN8 was found on B PLL, B non-Hodgkin lymphoma (NHL), and a subpopulation of normal B cells in the peripheral blood of healthy individuals [44]. The cryo-EM structure of a F_ab_ fragment of polatuzumab bound to IgM BCR was reported by Su et al. [17], and shows a stable complex is formed that causes little structural alteration to the IgM BCR. This was consistent with previous observations that polatuzumab binds to flexible sequences of the N terminus of CD79b [17,45]. After binding to CD79b, polatuzumab vedotin is internalized and releases its MMAE “payload” that induces apoptosis. However, binding to CD79b also induces AKT and ERK signaling that is associated with increased CD20 expression [46], which has potential benefits when treatment with polatuzumab vedotin is combined with CD20 monoclonal antibodies [46]. Pre-clinical data show polatuzumab vedotin is active against several diffuse large B cell (DLBCL) lines including activated B cell (ABC) and germinal center B cell like (GCB) subsets, and those with CD79b mutations that predominantly occur in ABC DLBCL patients who have poor survival [47].

In phase I and II trials, polatuzumab vedotin showed promising results in terms of safety, efficacy and survival outcomes as monotherapy in patients with relapsed/refractory (R/R) diffuse B cell lymphoma (DLBCL) [48,49], and in combination with rituximab in patients with R/R DLBCL or follicular lymphoma (FL) [50]. Polatuzumab vedotin in combination with bendamustine and rituximab (pola + BR) was then compared to BR in an open-label multicenter phase Ib/II trial for the treatment of R/R DLBCL or FL (Table 1) [51]. In DLBCL, pola + BR produced higher positron emission tomography (PET) complete response (CR) rates and longer progression free survival (PFS) and overall survival (OS) compared to BR regardless of prior treatment status. Common adverse reactions with pola + BR were neutropenia, thrombocytopenia, anemia, peripheral neuropathy, fatigue, diarrhea, pyrexia, decreased appetite and pneumonia, and cytopenias were the most common reason for discontinuing treatment (18% of all patients) [51]. Based on these results, multiple regulatory agencies including the United States Food and Drug Administration (FDA), the European Medicines Agency (EMA), and the British National Institute for Health and Care Excellence (NICE) granted approval for the use of polatuzumab vedotin (POLIVY, Genentech, Inc.) in combination with bendamustine and rituximab for the treatment of adult patients with R/R DLBCL after at least two prior therapies. In the most recent update, 192 patients in total had enrolled in the study (Table 1) [52]. A survival benefit with pola + BR versus BR persisted in the randomized arms and in the extension cohort, the CR rate was 38.7%; and PFS and OS were 6.6 months and 12.5 months, respectively [52].

Polatuzumab vedotin has entered phase III trials for various patient populations, including post-autologous stem cell transplant and in combination with CAR T cells, and conditions including mantle cell lymphoma, CLL, and Richter’s transformation. A phase III trial, the POLARIX study, that compared rituximab, cyclophosphamide, doxorubicin, vincristine and prednisone (R-CHOP) to pola-R-CHP in treatment naïve DLBCL patients, showed improved 2-year progression-free survival in the pola-R-CHP group (77% versus 70%; hazard ratio 0.73) (Table 1) [53]. However, to date, no difference in OS has been observed [53]. These results have led to the approval of pola-R-CHP for the treatment of adult patients with previously untreated DLBCL by the EMA, and the FDA will consider approval for this indication in 2023.

### 2.4. Potential BCR Targets Identified by Cryo-Electron Microscopy

The cryo-EM structure of IgM-BCR identified 3 epitopes that could be targeted for recognition by antibodies or mini-proteins (Figure 2) [17,73]. IgM-BCR is heavily glycosylated, however there are three surface sites that are free of glycosylation. These sites (Figure 2) are formed by the surfaces of the ECD of CD79a (ECDα) and one heavy chain Ig-like domain 4 (Cμ4_B_) (site 1), the surface of CD79b (ECDβ) and Cμ4_B_ (site 2), and the surface between ECDα and ECDβ that overlaps with site 1. Each of these sites contains hydrophilic surface loops that are antigenic [17], however due to potential steric hindrance would need to be targeted by small binders such as nano-bodies or mini-proteins [73,74]. Designing high-affinity peptides to target the BCR is a major challenge, however, there has been significant recent progress in designing binder proteins using deep learning methods. Using methods such as RoseTTAFold Diffusion [75], small peptides conjugated to an antitumor drug could be developed to target the surface sites to selectively deplete IgM positive B cells. Furthermore, a novel peptide could be fused to a Fc fragment for ligation of the IgM-BCR with its low affinity inhibitory receptor, FcγRIIb, which could be used in B lymphoproliferative diseases dependent on BCR signaling [17].

## 3. CD19, CD81 and CD21 Co-Receptor Complex

### 3.1. CD19 Regulation of the B Cell Receptor

BCR signaling occurs in association with a co-receptor complex made up of CD19, CD81 and CD21 [76]. As part of the complex, CD19 interacts directly with the BCR, increasing the response to antigen by lowering the BCR activation threshold [6], and amplifying the BCR signal by recruiting and activating Lyn, phosphoinositide 3-kinases (PI3Ks), and Btk [77,78,79]. CD19 influences cytoskeletal elements and the formation of BCR-antigen microclusters that form the fundamental B cell signaling unit [21]. As part of the complex, CD81 acts as a chaperone, binding CD19 in the endoplasmic reticulum and transporting it to the cell surface. On the surface, CD81 restricts the interaction between CD19 and the BCR that prevents amplification of constitutive tonic BCR signaling [18]. CD81 can switch between binding to CD19 and lipids and in high cholesterol environments such as lipid rafts, CD81 disengages from CD19 to preferentially bind to the surrounding lipid and interacts with the BCR [18]. The mechanism that changes CD81 from cholesterol free to cholesterol bound is not known but may be modulated by palmitoylation of CD81 [80].

CD21 forms part of a signaling complex with CD19 and CD81 [81], although it may also interact with CD35 when unable to associate with CD19 [82]. In mice, complement receptor type 2 enhances BCR activation [83], however in humans, co-clustering of CD21 and the BCR has been shown to reduce expression of activation markers, cytokine production, proliferation, and antibody production [29]. An anti-CD21 mAb has be shown to be a safe and relatively effective therapy for treatment of posttransplant B lymphoproliferative disorder [84], however CD21 as a therapeutic target has not resulted in any FDA approved mAbs and will not be discussed further in this review.

The cryo-EM structure of the CD19-CD81 complex was determined bound to the F_ab_ of the anti-CD19 monoclonal antibody, coltuximab (Figure 3) [18]. The Fab-bound CD19 ectodomain sits on top of the CD81 ectodomain and the transmembrane helices are arranged in a five-helix bundle.

Comparison of apo-CD81 without CD19 bound, as well as the CD19-CD81 complex, showed a large-scale conformational change within CD81 [18]. The two pairs of apo-CD81 TM helices form a cone that surrounds a large central cavity [85]. The large extracellular loop (EC2) made up of 5 helices contains a three-helix “stalk”. Following complex formation with CD19, the EC2 of CD81 transitions to an open conformation that together with movement of the EC2 helices and reorganization of the ectodomain, causes inward movement of the TM1-TM2 and TM3-TM4 pairs of helices, closing the central cavity and thus preventing cholesterol binding [85]. In the crystal structure of apo-CD81 [85], EC1 is disordered, though, following complex formation with CD19, EC2 interacts with EC1, stabilizing the open conformation of the ectodomain [18]. The ectodomain interface is hydrophobic, with several polar residues within hydrogen bonding distance providing a specificity contact site. Resolution of the structure of CD19-CD81 bound to coltuximab has confirmed that the anti-CD19 antibody, B43, and inebilizumab and denintuzumab bind to the same dominant epitope on the upper face of CD19 [18].

### 3.2. Targeting CD19 and CD81 for Treating B Cell Malignancies

CD19 is an attractive therapeutic target, being expressed on early B committed cells and is present on malignant B cells but is lost from plasma cells and not found on pluripotent stem cells and other healthy tissues [86]. While CD19 and CD20 are both highly expressed in B cell lymphomas, CD19 is more homogenously expressed and is maintained in CD20-negative tumor subsets and following anti-CD20 targeted therapy [87]. However, unlike CD20 antibodies, native CD19 antibodies are limited in their ability to elicit complement-dependent cytotoxicity (CDC), ADCC or ADCP [88]. Strategies to improve tumor cell killing by CD19 antibodies include antibody-drug conjugates (ADC) that deliver and release cytotoxic agents [89], T cell recruitment using the bispecific T cell engager (BiTE) molecule blinatumomab and CAR T cell products [90,91], and engineering the CD19 antibody F_c_ domain to overcome limitations of native CD19 antibodies (Table 1) [92].

### 3.3. Coltuximab Ravtansine

Coltuximab ravtansine (SAR3419) is a humanized anti-CD19 IgG1 mAb that is conjugated by a reducible linker to the tubulin maytansinoid inhibitor, DM4 [93]. Coltuximab binds to the upper face of CD19 (Figure 3), and it is likely that current therapeutic antibodies bind to the same dominant CD19 epitope. This is the same region as the anti-CD19 antibodies inebilizumab and denintuzumab mafodotin bind and the B43 antibody used in the bispecific T-cell engager (BiTE), blinatumomab [18]. Several phase I and II trials have shown coltuximab ravtansine to be a promising drug with acceptable safety and tolerability profiles (Table 1) [54,55,56].

### 3.4. Inebilizumab

Inebilizumab is a humanized, anti-CD19 mAb that targets and depletes CD19-expressing B cells via ADCC [94]. Inebilizumab was evaluated in a phase I dose-escalation study that included patients with R/R DLBCL, CLL, FL, or multiple myeloma who were ineligible for hematopoietic stem cell transplantation (Table 1) [57]. Although this study showed acceptable toxicity and preliminary efficacy in patients with R/R FL and DLBCL, further clinical evaluation in B cell malignancies has been discontinued. However, in June 2020, inebilizumab received global approval in the USA for the treatment of neuromyelitis optica spectrum disorder in adult patients who are seropositive for IgG autoantibodies against aquaporin-4 (AQP4-IgG) [94].

### 3.5. Denintuzumab Mafodotin (SGN-CD19A)

Denintuzumab mafodotin is an antibody-drug conjugate made up of a humanized anti-CD19 IgG1 mAb conjugated to monomethyl auristatin F (MMAF) [95]. After binding to CD19, the conjugate internalizes and releases MMAF, which binds to tubulin and induces apoptosis in the targeted cells. Promising outcomes were observed in a phase I trial in R/R B-NHL with overall response rate (ORR) 56% and CR rate 40% [96], however subsequent phase II trials showed unacceptable toxicities, with two studies (NCT02855359 and NCT02592876) terminated because of severe hematological adverse events.

### 3.6. Tafasitamab (XmAb5574)

Tafasitamab is a humanized anti-CD19 mAb with an engineered F_c_ domain that increases binding to F_c_ receptors on immune cells [88]. Increased F_c_-mediated effector functions allow tumor cell killing by ADCC, ADCP, and direct cytotoxicity [92,97]. Phase I studies demonstrated safety and efficacy data that supported phase II development of tafasitamab, and pre-clinical data suggested synergistic activity with lenalidomide against DLBCL (First–MIND study) [98]. Tafasitamab was then evaluated in a phase II single-arm, multi-center, open-label clinical trial (L-MIND) [58]. The primary endpoint was ORR (57%), and the median duration of response was 35 months (Table 1). The RE-MIND study was a matched comparison of data retrospectively collected from patients with R/R DLBCL treated with lenalidomide monotherapy for comparison with tafasitamab + lenalidomide-treated patients (L-MIND). The L-MIND and RE-MIND were cohorts of patients with ASCT-ineligible R/R DLBCL. The combination of tafasitamab and lenalidomide resulted in a better ORR, CR, and OS compared with lenalidomide monotherapy. Based on these studies, tafasitamab received accelerated approval from the US FDA in July 2020 for use in combination with lenalidomide to treat transplant-ineligible adults with R/R DLBCL [99]. Studies are ongoing to evaluate the use of tafasitamab as first-line treatment for DLBCL, First-MIND (NCT0413493657) and Front-MIND (NCT0482409258), and in R/R FL or marginal zone lymphoma (InMIND (NCT0468005259)).

### 3.7. Loncastuximab Tesirine

Loncastuximab tesirine is an ADC consisting of an anti-CD19 mAb conjugated to a cytotoxic DNA minor groove inter-strand cross-linking pyrrolobenzodiazepine dimer [100]. The LOTIS-2 trial, a phase II, single-arm study investigated loncastuximab tesirine in patients with R/R DLBCL or high-grade B-cell lymphoma who had failed at ≥two prior systemic regimens (Table 1) [59]. After a mean of 4.6 cycles, the ORR was 48%, and CR 25%. Median duration of response was 12.6 months for the 70 responders and not reached for patients with CR. For patients aged ≥75 years, with double-/triple-hit DLBCL or with transformed disease, ORRs were 52%, 33% and 45%, respectively [59].

### 3.8. Blinatumomab

The anti-CD19 antibody B43 is used together with anti-CD3, which is present on T cells, in the bispecific T-cell engager (BiTE) blinatumomab that targets CD19-positive B acute lymphoblastic leukemia (ALL) blasts and CLL cells [101,102,103]. The efficacy and safety of blinatumomab in the treatment of pre-treated Philadelphia chromosome (Ph)–negative R/R B cell precursor B ALL had been shown in single arm trials before a pivotal phase 2 trial led to accelerated FDA approval for the treatment of Ph-negative R/R B cell precursor ALL (Table 1) [60]. This approval was later expanded to Ph-positive B cell precursor ALL following publication of the TOWER study, a multinational, randomized, phase 3 trial that compared blinatumomab with standard chemotherapy in the treatment of patients with relapsed or refractory ALL (Table 1) [61]. This trial reported an improvement in OS for patients treated with blinatumomab compared to those treated with standard of care (hazard ratio = 0.71; 95% CI = 0.55, 0.93, *p* = 0.012). [61].

The crystal structure of B43 in complex with CD19 has been determined and shows B43 recognizing a conformational epitope made up of 3 loops in the middle portion of the ECD [104]. A comparison of B43 in the unbound form showed no changes in the quaternary structure of CD19, indicating a “lock-and-key” mechanism of binding. The central element of the epitope is a binding loop at residues 216–224, which in most non-human species has an insert that would disrupt binding. The presence of this insert explains the lack of cross-reactivity for B43 toward non-human species [104].

### 3.9. Antibodies against CD81

The CD81 antibody 5A6 recognizes a conformational epitope that is not accessible to binding when CD81 is complexed with CD19 but becomes accessible upon B cell activation [5]. In a xenograft model, 5A6 inhibits B cell lymphoma growth as effectively as rituximab, indicating possible therapeutic benefit [105]. Further studies are underway to test the safety and efficacy of 5A6 in animal models and eventually human trials [105].

## 4. CD22 Co-Receptor

### 4.1. CD22 Structure

CD22 is a member of the sialic acid binding immunoglobulin-like lectin (Siglec) surface membrane receptor family that binds to sialic acid containing glycans [106]. Siglecs are found on a wide range of immune cells including neutrophils, monocytes, dendritic cells, eosinophils, mast cells, T cells and B cells [106]. Depending on their cellular distribution and ligand specificity, they facilitate cell adhesion and/or cell signaling. The expression of CD22 is restricted to B cells where it has a critical role in establishing a baseline level of B cell inhibition [107]. CD22 is a transmembrane protein with an ECD made up of 7 immunoglobulin domains and an intracellular cytoplasmic tail [108]. Ligands bind to the extracellular N-terminus at the last immunoglobulin domain. CD22 binds to α2,6-linked sialic acid ligands linked to galactose that are expressed on several cell types including T and B cells [109]. CD22 can bind ligands on the surface of B cells in *cis*, or on the surface of other cells in *trans* configuration. However, *cis* interaction occurs most often with glycoprotein ligands on the same B cell [107]. CD22 exerts its inhibitory effect via phosphorylation of immunoreceptor tyrosine inhibitory motifs (ITIMs) in its cytoplasmic domain [110]. After antigen activation of the BCR, its associated CD22 molecule is phosphorylated and docking sites are formed for several SH2-domain-containing proteins, including the protein tyrosine phosphatase SHP-1 that dephosphorylates components of the BCR signaling cascade to dampen the BCR signal [111]. This role of CD22 in inhibiting BCR activation and its restricted expression to the B cell, makes CD22 a promising target for B cell depletion in B cell malignancies.

The CD22 ECD crystal structure was created using a truncated construct that contained the first 3 Ig domains (labelled d1 to d3) with 5 of the 6 N-linked glycosylation sites mutated to alanine [20]. The last immunoglobulin domain adopted the V-type fold that is responsible for binding α2,6-linked sialic acid ligands. The d2 domain contains two elongated beta strands that create a large interface with d1 and stabilize the orientation of the ligand binding domain [20]. The liganded and unliganded structures of CD22 are similar, which indicates carbohydrate binding by CD22 is primarily mediated by a preformed binding site [20]. The extracellular portion of CD22 adopts an extended conformation with low flexibility that is optimally configured to form nanoclusters and interact with self-ligands at the immune synapse [20].

The full-length CD22 ECD containing all 7 Ig domains, analyzed by negative-stain EM and small-angle X-ray scattering (SAXS), showed a limited range of conformations, again indicating low flexibility [20]. Furthermore, SAXS experiments of CD22 ECD in the presence of α2-6 sialyllactose showed no conformational change after ligand binding [20].

### 4.2. Antibodies against CD22

Studies of treatments targeting CD22 include mAbs, antibody-drug conjugates, radio-immunoconjugates, CAR-T cells, and bispecific antibodies [112]. To determine the epitope that epratuzumab binds to, a crystal structure of a glutamine resurfaced CD22 construct was complexed with epratuzumab F_ab_ and analyzed (Figure 4). This showed that epratuzumab primarily binds at the base of CD22 d2 but also interacts with d3 and that this site is not located at the ligand binding site [20]. The extensive N-linked glycosylation of CD22 is likely to affect how it can be targeted. The epratuzumab paratope includes an N-linked glycan at position N231, and a CD22 N231Q mutant that deletes this glycosylation site, results in a 6-fold improvement in binding affinity [20]. Furthermore, the binding affinity of epratuzumab to CD22 constructs was increased with reduced glycan size and content [20]. Therefore, it is likely that the degree of CD22 glycosylation affects the ability of epratuzumab to access its epitope.

### 4.3. Epratuzumab and Radioimmunoconjugates

Epratuzumab is a humanized IgG antibody that induces phosphorylation of CD22 and co-localization with SHP-1 in lipid rafts [113], leading to reduced Ca^2+^ flux after BCR stimulation [114]. Epratuzumab promotes movement of CD22 into lipid rafts and co-localization with CD79a that is then internalized together with CD22 [114]. This leads to decreased expression of the BCR complex on the B cell surface.

Naked and unconjugated epratuzumab has been studied in clinical trials in combination with chemotherapy or rituximab in NHL, including DLBCL, FL and small lymphocytic lymphoma [112], and B-ALL in adults and children [115,116]. However, epratuzumab has not been approved by the FDA for these conditions because of its limited efficacy and short duration of response.

Epratuzumab tetraxetan and BAY1862864 are radioimmunoconjugates that have been developed to target CD22 [65,117]. 90Yttrium -epratuzumab tetraxetan enables targeted radioisotope delivery and causes cell death via beta decay [117]. Epratuzumab tetraxetan has been studied as monotherapy in R/R NHL [62] and in combination with the anti-CD20 antibody veltuzumab in R/R aggressive NHL [63], as well as consolidation following RCHOP in DLBCL [64].

BAY1862864 is radiolabeled with 227-thorium that causes cell death and cell cycle arrest via DNA damage from alpha particle emission [65]. The first in-human study was in R/R NHL. Despite its small sample size, disease heterogeneity and early termination of the trial due to patients with progression enrolling into other clinical trials, which prevented the evaluation of efficacy, BAY1862864 was safe and well-tolerated [65].

### 4.4. CD22 Antibody-Drug Conjugates

Several ADCs targeting CD22 have been developed, including inotuzumab ozogamicin, FDA approved for use in relapsed/refractory B-ALL, and moxetumomab pasudotox-tdfk, approved for use in patients with R/R hairy cell leukemia (HCL) after two prior lines of therapy. Inotuzumab ozogamicin is a human anti-CD22 antibody attached to calicheamicin, a DNA-damaging agent that leads to apoptosis when intracellularly activated [118]. The FDA approved inotuzumab ozogamicin based on data from the INO-VATE ALL study that compared inotuzumab ozogamicin monotherapy to standard of care intensive chemotherapy [66,67]. At final analysis, inotuzumab ozogamicin had superior OS and PFS compared to standard of care chemotherapy (Table 1) [66]. Veno-occlusive disease (VOD) occurred in 14% of patients receiving inotuzumab ozogamicin, and risk factors for the development of VOD should be considered before therapy. Inotuzumab ozogamicin has been studied in first relapsed B-ALL in combination with lower-intensity chemotherapy and blinatumomab [68], as first line therapy [69], and is an effective treatment option in these settings (Table 1). Several trials have evaluated inotuzumab ozogamicin in NHL but with less promising results compared to B-ALL [119,120,121].

Moxetumomab pasudotox-tdfk is a recombinant immunotoxin consisting of a fragment variable (Fv) of murine CD22 monoclonal antibody fused to a truncated portion of Pseudomonas exotoxin A [122]. In the pivotal trial, patients with R/R HCL who received moxetumomab pasudotox for up to 6 cycles had high rates of durable responses and minimal residual disease eradication with acceptable tolerability [70].

### 4.5. CD22 CAR-T Therapies

Patients with B-ALL who are resistant to or relapse after CD19-targeted immunotherapies have few treatment options, however CARs targeting CD22 have shown promising efficacy and safety in early phase trials for several B cell cancers [71].

A CD22 CAR T cell has been tested for safety in a phase I trial with R/R B-ALL, with 80% previously receiving CD19 targeted therapy (Table 1) [71]. Dose dependent anti-leukemic activity was observed and established the clinical activity of CD22 CAR T cells in pre-B ALL [71]. In another phase I trial (Table 1), CD22 CAR T-cell therapy was evaluated in 58 patients with relapsed/refractory CD22+ malignancies (56 with B-ALL, one DLBCL and one chronic myeloid leukemia with ALL blast crisis). Most patients had undergone transplant (67%) or CD19 CAR T-cell therapy (62%). The CR rate was 70% and enabled bridging to allogeneic hematopoietic stem cell transplant in 13 patients [73]. In both CD22 CAR T-cell studies, most patients who relapsed had CD22-negative or -dim disease at relapse. To prevent release by antigen escape, bispecific CD20/CD22 CARs, CD19/CD22 CARs and tri-specific CD19/CD20/CD22 CAR-T cell are in development [123,124].

## 5. Conclusions

The development of mAbs that bind to known B cell surface antigens has revolutionized the treatment of B cell cancers. In addition to killing malignant B cells directly by ADCC, mAbs against B cell antigens have been conjugated to cytotoxins and radioisotopes [89], combined with other antibodies that bring the tumor cell in contact with T cells [12,13], and used in CAR T and NK cells to target malignant B cells [89,123,124]. The CD20 mAb—rituximab—was approved by the FDA in 1997 to treat B cell non-Hodgkin lymphoma, and since then its use has expanded to other B cell malignancies and autoimmune diseases. However, CD20 is not expressed by all B cell malignancies and can be downregulated by cells to escape CD20 antibody immunotherapies. Consequently, engineered antibodies have been developed against other B cell antigens, including epitopes expressed by CD79b as part of the BCR, and the BCR co-receptor molecules, CD19, CD81 and CD22 (Figure 1).

A successful therapeutic mAb for the treatment of B cell cancers has high specificity for its target whose expression should be restricted to malignant B cells. CD79b is expressed in early B cell development and is only lost in the late plasma cell stage, although it is found in some myelomas [41]. CD19 is expressed on early committed B cells and malignant B cells but not on pluripotent stem cells and other healthy tissues [86], and its expression is maintained after CD20 antibody targeted therapy [87]. The expression of CD22 is restricted to B cells and it exerts an inhibitory effect on BCR signaling after antigen activation [110,111]. This role of CD22 in inhibiting BCR activation and its restricted expression to the B cell, makes it a promising target for B cell depletion in B cell malignancies. Several mAbs targeting BCR-associated proteins have been developed and approved for use by the FDA in B cell malignancies and autoimmune diseases. However, until recently, high-definition models of these proteins have not been available for structure-guided engineering of mAbs.

The cryo-EM structure of IgM-BCR, in addition to establishing the binding site of polatuzumab, has identified three epitopes that could be targeted for recognition by antibodies or miniproteins (Figure 2) [17,73]. Each of these sites contains antigenic hydrophilic surface loops, however it is only highly detailed structural information that has identified the requirement for targeting by small molecules such as nanobodies or miniproteins [73,74].

The CD19-CD81 complex was identified bound to the anti-CD19 mAb, coltuximab [18]. Coltuximab binds to the upper face of CD19, which is the same region as the anti-CD19 antibodies inebilizumab and denintuzumab mafodotin bind and the B43 antibody used in blinatumomab [18]. Following the binding of antigen to the BCR, CD19 moves away from CD81, inducing a large conformational change within CD81 [18]. The CD81 antibody, 5A6, recognizes an epitope that is not accessible when CD81 is complexed with CD19 but becomes accessible upon B cell activation [5]. This structural information will be critical to future development of CD81 mAbs.

The structures at the interface between binding partners can be determined by X-ray, cryo-EM, cross-linking mass spectrometry, protein knock-out and other techniques. The converse where an “antigenic site” is identified and used as a target for making mAbs is far more challenging. However, the recently reported structures of the IgM- and IgG-BCRs, and the BCR co-receptors CD19, CD81 and CD22 will help guide the rational design of antibodies, nanobodies, and miniproteins for the treatment of B cell malignancies and autoimmune diseases.

## Figures and Tables

**Figure 1 cancers-15-02881-f001:**
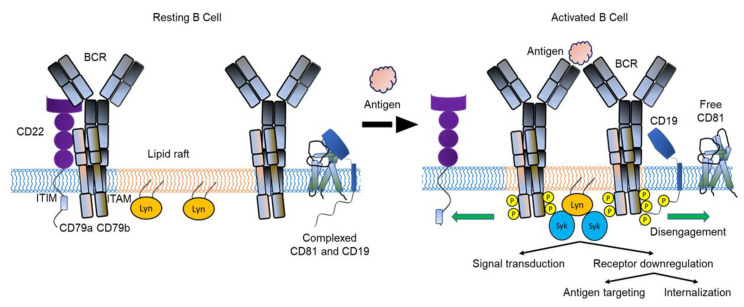
Model of BCR dynamics during B cell activation. Antigen binding to the BCR promotes the interaction between BCRs and lipid rafts. The BCR moves away from the negative co-receptor CD22 and the CD19-CD81 complex dissociates, allowing CD19 to diffuse in the membrane and interact with the BCR. Interaction with CD19 amplifies signaling through the BCR and activation of the B cell. CD22 is not shown here as part of the CD19-CD81 complex. BCR, B cell receptor; ITAM, immunoreceptor tyrosine-based activation motifs; ITIM, immunoreceptor tyrosine-based inhibition motif; Lyn, Lck/Yes-related novel (Lyn) tyrosine kinase; P, phosphate; Syk, spleen tyrosine kinase. Modified from Susa et al. [5].

**Figure 2 cancers-15-02881-f002:**
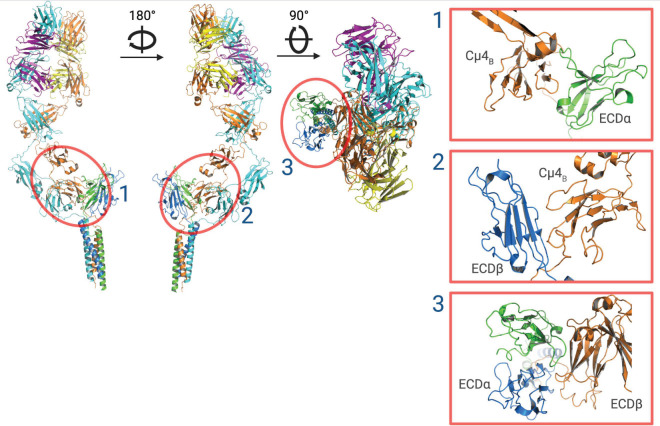
Ribbon structure of the IgM-BCR. Cryo-EM reconstruction of the IgM-BCR showing CD79a (green), CD79b (blue), IgM heavy chains (orange and cyan) and light chains (magenta and yellow). Using the cryo-EM structure of IgM-BCR, three surface (1–3) epitopes that are free of glycosylation sites are identified for recognition by an antibody. The glycan-free surface epitopes are indicated by red circles and magnified views are shown in the right-side panels. ECDα structural model produced using PyMOL software version 2.5.2, Schrödinger, Inc., New York, NY, USA [35].

**Figure 3 cancers-15-02881-f003:**
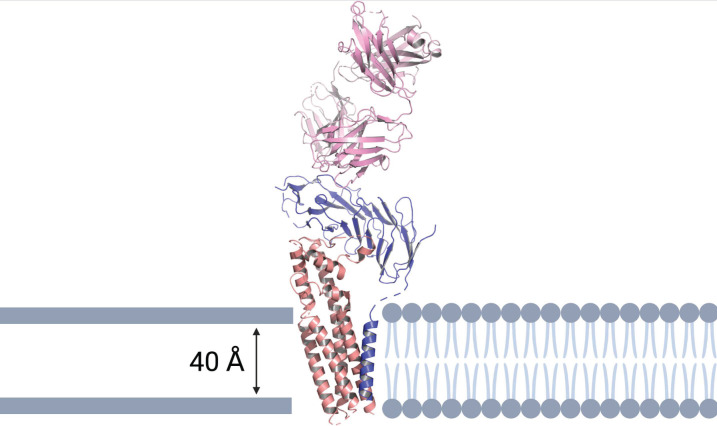
Ribbon representation of the human CD19-CD81 complex within a cell membrane bound to coltuximab. CD19 is colored blue, CD81 tan, coltuximab pink, and phospholipids in grey cartoon representation. The model shows the heavy and light chains of coltuximab (pink), the complete ECD and TM of CD19 (blue), and the full-length of CD81 (tan). The structural model was produced using PyMOL software version 2.5.2., Schrödinger, Inc., New York, NY, USA [35].

**Figure 4 cancers-15-02881-f004:**
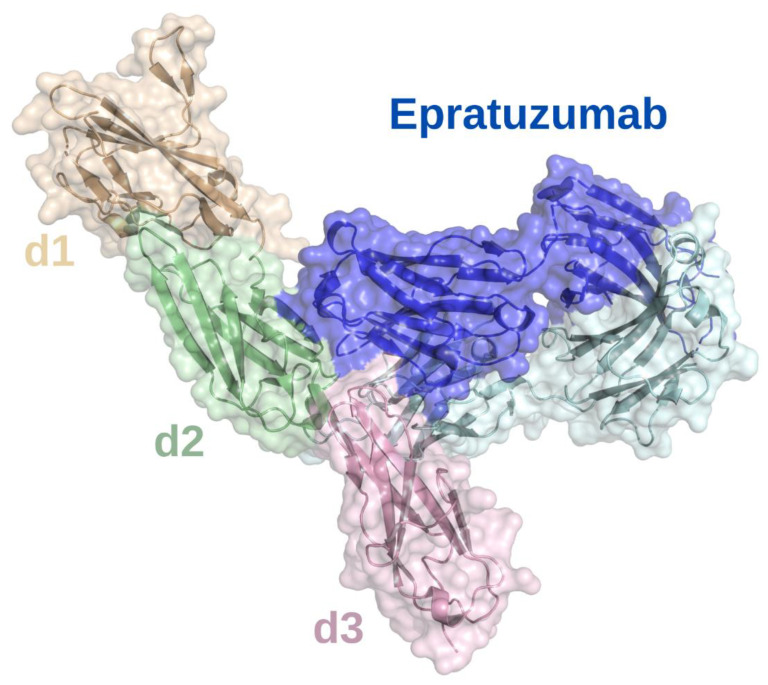
Surface epitope of CD22 recognized by epratuzumab. X-ray crystallographic structure of epratuzumab (blue and cyan) bound to the CD22 d2 (pale green)/d3 (lilac) interface. The CD22 d1 Ig domain is shown in sand. The structural model was produced using PyMOL software version 2.5.2, Schrödinger, Inc., New York, NY, USA [35].

**Table 1 cancers-15-02881-t001:** Selected monoclonal antibody-based therapies targeting CD79a, CD19 and CD22 in B cell malignancies.

Monoclonal Antibody	Characteristics	Selected Studies	Disease	Outcomes
Anti CD79b				
Polatuzumab vedotin	CD79b-directed antibody covalently linked to the anti-mitotic cytotoxic agent monomethyl auristatin (MMAE)	Phase Ib/II multicenter trial [51,52]	R/R DLBCL or FL	Pola + BR: ORR was 63% compared to 25% for BR alone. Pola + BR: CR rate was 40% compared to 18% for BR alone.
		Phase III trial [53]. R-CHOP versus pola-R-CHP	Treatment naïve DLBCL	No difference in OS to date.
Anti CD19				
Coltuximab ravtansine (SAR3419)	Humanized IgG1 anti-CD19 monoclonal antibody conjugated to the maytansinoid tubulin inhibitor, DM4	Phase I multidose-escalation study [54]	R/R DLBCL, some previously treated with rituximab-based therapy	ORR 23.5% (*n* = 17)
		Phase II, single-arm, multicenter study [55]	R/R DLBCL previously treated with rituximab-based therapy	ORR was 44% and CR rate 15% (*n* = 41)
		Phase II, single-arm, multicenter study combined with rituximab [56]	R/R DLBCL	ORR was 31% and CR rate 8.9% (*n* = 45)
Inebilizumab	Humanized, anti-CD19 monoclonal antibody	Phase I dose-escalation study [57]	R/R DLBCL, CLL, FL, and multiple myeloma	ORR was 60% (*n* = 20)
Tafasitamab (XmAb5574)	Humanized anti-CD19 monoclonal antibody with an engineered Fc domain that increases binding to Fc receptors	Phase II single-arm, multi-center, open-label study in combination with lenalidomide [58]	R/R DLBCL	ORR was 57%.
Loncastuximab tesirine	Anti-CD19 mAb conjugated to a cytotoxic DNA minor groove inter-strand cross-linking pyrrolobenzodiazepine dimer	Phase II, single-arm study [59]	R/R DLBCL or high-grade B-cell lymphoma	ORR was 48% and CR rate 25% (*n* = 145) Median duration of response was 12.6 months for responders
Blinatumomab	A bispecific T-cell engager (BiTE) that combines anti-CD19 antibody B43 with anti-CD3	Multicenter, single-arm, open-label phase II study [60].	Philadelphia-chromosome-negative, primary refractory or relapsed B cell precursor ALL	CR with complete or partial hematologic recovery of 43%, and median overall survival of 6.1 months
		Multinational, randomized, phase III trial that compared blinatumomab with standard of care chemotherapy [61]	R/R ALL	Median OS was 7.7 months in the blinatumomab arm (95% CI = 5.6, 9.6) and 4 months in the standard of care chemotherapy arm (95% CI = 2.9, 5.3)
Anti-CD22				
Epratuzumab tetraxetan	^90^yttrium-labeled humanized IgG anti-CD22 antibody	Multicenter, phase I/II study of epratuzumab tetraxetan alone [62]	R/R NHL	ORR was 62%, CR rate 48%, and median PFS was 9.5 months
		In combination with the anti-CD20 antibody veltuzumab [63]	R/R NHL	ORR was 53%, CR rate was 18%, and PR rate was 35%
		single-group, phase II trial epratuzumab tetraxetan as consolidation after first-line induction chemoimmunotherapy [64]	Untreated elderly patients with DLBCL.	Estimated 2 year event-free survival was 75% (95% CI 63–84)
BAY1862864	(227) thorium-labeled epratuzumab	dose-escalation phase I study [65]	R/R NHL	ORR of 38%, PR of 19% and CR of 5% (*n* = 21)
Inotuzumab ozogamicin	Anti-CD22 antibody attached to calicheamicin	Randomized, phase III INO-VATE study. Inotuzumab versus standard-of-care chemotherapy [66,67]	R/R B cell precursor ALL	Patients treated with inotuzumab ozogamicin had higher CR rates (74% versus 35%), and MRD-negativity (78% versus 28%). The inotuzumab arm had longer PFS (HR, 0.45; 97.5% CI, 0.34–0.61; *p* < 0.001) and OS (HR, 0.77; 97.5% CI, 0.58–1.03; *p* = 0.04).
		Phase II study in combination with lower-intensity chemotherapy and blinatumomab [68]	First relapsed Philadelphia chromosome-negative B-ALL	ORR was 92% and CR 73% (*n* = 44)
		Single-arm, phase II study, Inotuzumab ozogamicin in combination with low-intensity chemotherapy [69]	Older patients with Philadelphia chromosome-negative ALL	With a median follow-up of 29 months, 2-year PFS was 59%.
Moxetumomab pasudotox-tdfk	Fv of murine CD22 mAb fused to a truncated portion of Pseudomonas exotoxin A	Multicenter, open-label study [70]	R/R HCL	CR rate was 41% and ORR was 75%
CD22 CAR-T therapies	Chimeric antigen receptor T cells targeting CD22	Phase I, first-in-human, dose escalation trial [71]	R/R CD22 expressing hematopoietic malignancies	CR in 73% (11/15) of patients receiving ≥1 × 106 CD22 CAR T cells/kg
		Single-center, phase I, 3 1 3 dose-escalation trial [72]	R/R CD22 expressing hematopoietic malignancies	CR was 70% and median OS was 13.4 months (*n* = 58)

Abbreviations: ALL, acute lymphoblastic leukemia; ASCT, autologous stem cell transplant; BR, bendamustine and rituximab; CAR, chimeric antigen receptor; CLL, chronic lymphocytic leukemia; CR, complete response; DLBCL, diffuse large B cell lymphoma; FL, follicular lymphoma; HCL, hairy cell leukemia; MRD, minimal residual disease; NHL, non-Hodgkin lymphoma; ORR, overall response rate; OS, overall survival; PFS, progression free survival; Pola, polatuzumab vedotin; R/R, relapsed or refractory.

## Data Availability

Not applicable.

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
