# Peer review of "Combination of High-Resolution Structures for the B Cell Receptor and Co-Receptors Provides an Understanding of Their Interactions with Therapeutic Antibodies"

_cancers, 2023, doi:10.3390/cancers15112881_

Round 1

Reviewer 1 Report

Bhattacharyya et al provide a well written and comprehensive review of the structure and function of components of the B cell receptor (BCR) and an overview of BCR-targeted therapeutics, primarily antibody-based. I have only minor comments.

·         Although the title stresses the use of high-resolution structures to help guide development of such therapeutics, this is not really discussed. For example, in section 2.4., it would be good to add some commentary around how novel IgM BCR epitopes revealed by cryo-EM structures may help guide development of improved therapeutics. Similarly, in section 3, it would be good to discuss how knowledge of CD19 structure in complex with antibody has helped understanding of MOA and/or may help guide therapeutic development.

·         Line 70; monoclonal antibodies can also induce phagocytosis (ADCP) and direct cell killing – for example see work by Mark Cragg and others on CD20 antibodies.

·         First sentence paragraph 2.2 is a little misleading: The antibodies discussed  are not tumour specific they are B cell specific.

Reviewer 2 Report

This is a timely and well-written article combining recently obtained insights into the architecture of the BCR with new perspectives of designing antibody-based tools to interfere with B cell malignancies. Also, the concise summary of BCR signaling events is appreciated. Hence, this review will be of interest for the community of basic researchers and clinicians alike. 

I have only minor comments/recommendations.

- Firstly, authors describe antigen-induced BCR signaling as compared to "tonic" BCR signaling, but the latter appears a little bit under-represented. I am aware that the nature of tonic BCR signaling is still mysterious (even though PI3K has been suggested to play a role). However, the role of tonic and chronic BCR signaling for the survival of certain B lymphoma cells could be a bit more detailed.

- Secondly, authors carefully distinguish between the various BCR isotypes. However, the role of the IgG-BCR class and its unique signaling properties as a prototypic memory type BCR appears should be discussed.

- Lastly, given the key role of Ig-alpha and Ig-beta and their ITAM- and non-ITAM based signaling elements, citing the original publications would be appropriate.
